# Identification of Salt Tolerance Related Candidate Genes in ‘Sea Rice 86’ at the Seedling and Reproductive Stages Using QTL-Seq and BSA-Seq

**DOI:** 10.3390/genes14020458

**Published:** 2023-02-10

**Authors:** Qinmei Gao, Hongyan Wang, Xiaolin Yin, Feng Wang, Shuchang Hu, Weihao Liu, Liangbi Chen, Xiaojun Dai, Manzhong Liang

**Affiliations:** Hunan Province Key Laboratory of Crop Sterile Germplasm Resource Innovation and Application, Hunan Normal University, Changsha 410081, China

**Keywords:** SR86, salt tolerance, bulked segregant analysis (BSA), salt-tolerant QTL

## Abstract

Salt stress seriously affects plant growth and development and reduces the yield of rice. Therefore, the development of salt-tolerant high-yielding rice cultivars through quantitative trait locus (QTL) identification and bulked segregant analysis (BSA) is the main focus of molecular breeding projects. In this study, sea rice (SR86) showed greater salt tolerance than conventional rice. Under salt stress, the cell membrane and chlorophyll were more stable and the antioxidant enzyme activity was higher in SR86 than in conventional rice. Thirty extremely salt-tolerant plants and thirty extremely salt-sensitive plants were selected from the F_2_ progenies of SR86 × *Nipponbare* (Nip) and SR86 × 9311 crosses during the whole vegetative and reproductive growth period and mixed bulks were generated. Eleven salt tolerance related candidate genes were located using QTL-seq together with BSA. Real time quantitative PCR (RT-qPCR) analysis showed that *LOC_Os04g03320.1* and *BGIOSGA019540* were expressed at higher levels in the SR86 plants than in Nip and 9311 plants, suggesting that these genes are critical for the salt tolerance of SR86. The QTLs identified using this method could be effectively utilized in future salt tolerance breeding programs, providing important theoretical significance and application value for rice salt tolerance breeding.

## 1. Introduction

Salt stress is one of the main factors affecting crop production [1,2]. Excess amounts of salt in the soil affect plant photosynthesis, metabolism, and respiration, leading to stunted growth, reduced grain yield, and eventually plant death. About six percent of the land area worldwide is affected by salinization [3]. A low salt concentration can significantly reduce the yield of many rice (*Oryza sativa* L.) varieties. Therefore, improving the ability of rice to tolerate salt will assist in making the full use of land resources and improve agricultural production and crop yield [4].

The characteristic of salt tolerance in rice is controlled by multiple genes [5,6], some of which have been studied using different approaches including mapping-by-sequencing (MBS), QTL-sequencing (QTL-seq), and RNA-sequencing (RNA-seq). With the emergence of new high-throughput sequencing technologies, such as whole genome re-sequencing, BSA-seq, and RNA-seq, quantitative trait loci (QTL) identification has become highly effective in analyzing the complex trait of salt tolerance in plants [7]. Multiple salt tolerance related genomic regions have been identified through QTL-seq in many different plant species to date, such as Arabidopsis [8], soybean [9], wheat [10], and sorghum [11]. Many salt tolerance QTLs have been identified in rice at the bud stage and some related genes have also been cloned and functionally characterized [12,13,14,15]. In addition, some new salt tolerance related QTLs have also been identified at the heading and reproductive growth stages [16,17,18,19]. Deciphering the physiological and biochemical reactions of salt-sensitive and salt-tolerant rice genotypes is extremely important for improving plant salt tolerance. Considerable study has been performed on QTL mapping in rice; however, the availability of rice germplasm resources with strong salt tolerance is lacking, and only a few QTLs have been applied to plant breeding to date. The identification of highly salt-tolerant germplasm resources, exploration of their salt tolerance related QTLs or genes, and application of these QTLs or genes for the generation and cultivation of new salt-tolerant varieties are tasks of great significance. SR86 is a semiwild salt-tolerant rice germplasm found submerged in sea water in 1986 [20]. In recent years, some progress has been made in the research of salt tolerance related QTLs in SR86. The transcriptome analysis of SR86 revealed lots of differentially expressed genes (DEGs) under NaCl treatment conditions [21]. The genome-wide association study (GWAS) of SR86 under different salt concentrations led to the identification of 51 loci significantly related to salt stress [22]. However, only a few reports are available on the physiological mechanism of salt tolerance and the mapping of salt tolerance related genes in SR86.

In this study, the physiological and biochemical mechanisms of salt tolerance in SR86 were studied at the seedling, tillering, and reproductive stages. A total of two F_2_ populations were produced by crossing the salt-tolerant cultivar SR86 with the *japonica* rice variety *Nipponbare* (Nip) and *indica* rice variety 9311. Salt-tolerant QTLs were identified during the whole growth cycle and at the reproductive growth stage by BSA and QTL mapping. Real time quantitative PCR (RT-qPCR) was used to identify the salt-tolerant candidate gene of SR86. These candidate genes could be used to explore the molecular mechanisms of salt tolerance, improve genetic resources, and facilitate salt tolerance breeding in rice.

## 2. Materials and Methods

### 2.1. Experimental Material

SR86, a rice germplasm first discovered in 1986 by Chen et al. [23] at Zhanjiang (Guangdong, China) that is capable of growing in saline-alkaline soil, was provided by the rice germplasm resource platform of the Hunan Rice Research Institute. Detailed information of the cultivar can be found at https://www.ricedata.cn/variety/varis/618124.htm (accessed on 8 June 2018). The *japonica* and *indica* rice cultivars (Nip and 9311, respectively) were obtained from the Hunan Province Key Laboratory of Crop Sterile Germplasm Resource Innovation and Application, China. The F_2_ populations used in the research were derived from the cross of SR86 (donor genotype) with *Nipponbare* (Nip) and 9311 (recipient genotypes). All materials were planted in the experimental fields of Hunan Normal University, Changsha, Hunan Province, China.

### 2.2. Measurement of Phenotypic Traits

The germinated seeds were planted in a normal hydroponic medium. Rice seedlings were grown to the three-leaf stage (approximately 14 days) in normal (NaCl-free) hydroponic medium and then transferred to 150 mM NaCl for 4 days. To facilitate the recovery, the plants were returned to the normal hydroponic medium and grown for 5 days [24]. Finally, the chlorophyll content of the leaves was measured.

The phenotypic traits of parental lines and offspring were evaluated in greenhouses under natural illumination at 25–28 °C [25]. The germinated seeds were planted in plastic seedling pots for growth and normal seedling management. When the plants grew to the three leaf and four tiller stage, the soil was dried under natural conditions and then treated with a 0.9% salt (*w*/*v*) solution, with the water level at 1.5 cm above the soil. Finally, the phenotypic data were measured and statistically analyzed.

### 2.3. Measurement of Chlorophyll and Proline Contents

To measure the chlorophyll content of the rice plants, chlorophyll was extracted from ground fresh leaves (100 mg) and the absorbance of each sample was tested at 663 and 645 nm using a UV2400 UV/VIS spectrophotometer [26].

The content of proline was measured using the Nanjing Jiancheng kit. Briefly, 1 mL of proline extract was added to 0.1 g of leaves and the sample was incubated in an ice bath for homogenization. Subsequently, the sample was incubated in a boiling water bath for 10 min and then centrifuged at 10,000× *g* for 10 min at room temperature. The supernatant was removed and cooled and its absorbance was measured at 520 nm.

### 2.4. Measurement of Electrical Conductivity and Malondialdehyde (MDA) Content

Leaves were rinsed with deionized water and soaked dry. Then, 25 mL of deionized water was added to the test tube containing the leaves, and the samples were incubated at 25 °C for 1 h. The electrical conductivity of the leachate was measured before and after the salt treatment (*a*1 and *a*2, respectively) with a conductivity meter. Finally, the samples were placed in a boiling water bath for 15 min, and the electrical conductivity after boiling (absolute electrical conductivity; *a*3) was measured. The relative electrical conductivity (%), an indicator of cell membrane permeability, was calculated for each sample using the equation below:$$\mathrm{Relative}\;\mathrm{electrical}\;\mathrm{conductivity}=\frac{\left(a2-a1\right)}{\left(a3-a1\right)}\times 100$$

The degree of lipid peroxidation was determined by measuring the MDA content, as described previously [24]. Fresh leaves were homogenized in trichloroacetic acid (TCA) on ice and the supernatant was added to thiobarbituric acid (TBA). Samples were incubated in a boiling water bath for 15 min and the absorbance was measured at 450, 532, and 600 nm after cooling to room temperature.

### 2.5. Evaluation of Antioxidant Enzyme Activity

The activities of superoxide dismutase (SOD), peroxidase (POD), and catalase (CAT) enzymes were evaluated according to the corresponding kit instructions (Jiancheng Bioengineering Institute, Nanjing, China), according to the manufacturer’s instructions. Plant tissue was ground in nine volumes of phosphoric acid buffer (pH 7.4) (1 g tissue: 9 mL buffer) in an ice bath. Then, the sample was centrifuged at 3500 rpm for 10 min and the absorbance of the supernatant was measured at 420, 550, and 595 nm.

### 2.6. Salt Stress Treatments

F_2_ seeds of the SR86 × Nip cross were cultured in a light incubator at 37 °C for 3 days. Seeds showing uniform growth were sown in the nursery. At the three-leaf stage, the seedings were transplanted into soil containing 150 mM NaCl. The phenotypes of rice seedlings under salt stress conditions (150 mM NaCl) were observed and recorded. Thirty extremely salt-sensitive plants (the earliest wilted plant) and thirty extremely salt-tolerant plants, as well as plants of the two parental genotypes (SR86 and Nip), were selected at the seedling and reproductive stages, immediately frozen in liquid nitrogen, and then stored at −80 °C. Four mixed DNA pools (R01-SR86, R02-Nip, R03-salt-tolerant, R04-salt-intolerance) were constructed and BSA was performed by Biomarker Biotechnology Co., Ltd. (Beijing, China).

The SR86 × 9311 F_2_ seeds were cultured in a light incubator at 37 °C for 3 days. The seeds showing uniform growth were sown in the nursery. A total of 50 parental and 616 F_2_ seedlings, each with three to four tillers, were numbered. The tillers of each seedling were separated to obtain two seedlings. One of these two seedlings was transplanted into the soil containing no salt and the other was transplanted into soil containing 150 mM NaCl. The phenotypes of the rice seedlings under salt stress conditions (150 mM NaCl) were observed and recorded during the whole growth period. Thirty extremely salt-sensitive plants (shortest life cycle) and thirty extremely salt-tolerant plants, as well as plants of the two parental lines (SR86 and 9311), were selected from the control (0 mM NaCl) treatment at the seedling and reproductive stages, immediately frozen in liquid nitrogen, and then stored at −80 °C. Four mixed DNA pools (SR86, 9311, G1-salt-tolerant, G2- salt-intolerance) were constructed and BSA was performed by Novogene (Beijing, China).

### 2.7. BSA

DNA was extracted from each plant in the two selected salt-tolerant and salt-intolerant population, and the quality and concentration of the extracted DNA were tested by the Nanodrop software. An equal amount of DNA was taken to construct the extreme phenotype mixing pool, i.e., the salt-tolerant pool and salt-intolerant pool. Sequencing was performed using Illumina HiSeq. The sequences from the mixing pool measurements were compared to the Nip genome and SNP and Indel annotations were performed using Illumina Casava 1.8. High-quality SNPs were obtained by filtering SNPs from the population of SR86/Nip. The threshold was calculated by the Euclidean Distance (ED) algorithm. The location interval was determined by the association threshold and the results were calculated by the SNP-index method. Based on the SNP, the results of the two association analysis methods were overlapped. The Indel was filtered to obtain high quality credible INDEL sites and the threshold was calculated by the Euclidean Distance (ED) algorithm. The location interval was determined by the association threshold and the results were calculated by the SNP-index method. Based on the Indel, the intersection of the results of the two association analysis methods was obtained.

DNA was extracted from each plant in the two selected salt-tolerant and salt-intolerant populations and the quality and concentration of the extracted DNA were tested by the Nanodrop software. An equal amount of DNA was taken to construct the extreme phenotype mixing pool, i.e., the salt-tolerant pool and salt-intolerant pool. Valid sequencing data were aligned to the reference genome (9311) by the BWA software (parameters: mem-t 4-k 32-m). The results needed to be checked by the SAMTOOLS software to remove duplicates. The mutation was annotated and predicted by the SNPEFF software. Based on the results of genotyping, polymorphic markers were screened for homozygous differences between the two parents and 9311 was selected as the reference parent. The SNP-index (frequency of SNP) of marker sites between the parents was calculated for two progenies. In order to reflect the distribution of the SNP-index on the chromosome directly, the distribution of the SNP-index on the chromosome was plotted. The mean of the SNP-index in each window was calculated to reflect the SNP-index distribution of the offspring. Then △(SNP-index) was calculated. The Indel-index was analyzed in the same way as the SNP-index. In order to reflect the distribution of the All-index on the chromosome after the SNP-index and Indel-index were combined, the distribution of the All-index on the chromosome was plotted. The window that was greater than the threshold as the candidate interval was selected. A window larger than the threshold was used as the candidate interval.

### 2.8. Gene-Ontology Analysis

Gene-ontology analysis was conducted using the cluster Profiler, referring to the Ashburner et al. approach [27].

### 2.9. RT-qPCR Analysis

The total RNA was extracted from Nip plants using the TRIzol Reagent (Invitrogen, Waltham, MA, USA) and reverse transcribed using the reverse transcription kit. The resultant cDNA was subjected to quantitative analysis using the Takara Quantitative Kit RR420A. Then, RT-qPCR was performed on the ABI PRISM 7500 Real time PCR instrument (Applied Biosystems, Waltham, MA, USA). The following system was used: 2 × SYBR Premix Ex Taq 10 μL, Forward primer 0.8 μL, Reverse primer 0.8 μL, cDNA template 0.5 μL, ddH_2_O up to 20 μL. The reaction procedures were: 94 °C for 30 s; 94 °C for 5 s, 55 °C for 15 s, 72 °C for 10 s, 40 cycles, 72 °C for 1 min. Dissolution curves were set from 65.0 °C to 95.0 °C for 5 s, with intervals of 0.5 °C. Each reaction was set to 3 replicates. The data were analyzed by 2^−ΔCT^ and 2^−ΔΔCT^ methods. The primers used for RT-qPCR were designed using Primer 5 (Appendix A) and synthesized by Sangon Biotech. The actin gene of rice was used as an internal reference.

### 2.10. Data Analysis

The mean ± standard deviation (SD) of three independent replicates was calculated. The average of three replicates for each treatment was calculated using the PASW statistics18 software and statistically significant differences were detected at *p* < 0.05 using one-way analysis of variance (ANOVA) and Student’s *t*-test. Figures were constructed using a GraphPad Prism5.

## 3. Results

### 3.1. Phenotypic Analysis of Salt Tolerance in SR86

SR86 is a highly salt-tolerant rice germplasm resource originally identified in China [20]. We compared the salt tolerance of SR86 with that of the *japonica* rice Nip and *indica* rice 9311 grown in 150 mM NaCl. SR86 showed a greater salt tolerance (Figure 1a) and significantly higher survival rate and chlorophyll content (Figure 1b,f) compared with Nip and 9311. 

To further study the molecular mechanism of salt tolerance in SR86, we measured various physiological and biochemical parameters of SR86, Nip, and 9311 plants under salt stress conditions. No significant differences were observed among plants of the three cultivars under normal conditions. However, under salt stress conditions, the accumulation of lipid peroxidation products, as indicated by the measurement of electrical conductivity, and MDA in SR86 was significantly lower than that in Nip and 9311 (Figure 1d,e), while the content of proline and the activities of various antioxidant enzymes (SOD, POD, and CAT) were significantly higher in SR86 (Figure 1c,g–i). Overall, the stability of the cell membrane and chlorophyll content, the activity of antioxidant enzymes, and the accumulation of osmotic substances were greater in SR86 than in Nip and 9311 plants under salt stress conditions.

To further understand the relationship between these physiological and biochemical parameters, a correlation analysis was performed. The results showed that chlorophyll content was negatively correlated with electrical conductivity and MDA; proline content was positively correlated with SOD, POD, and CAT; electrical conductivity was positively correlated with MDA; and antioxidant enzyme activity was positively correlated (Figure 2). These results showed that salt tolerance was positively correlated with the chlorophyll content, proline content, and antioxidant enzyme activity and negatively correlated with the electrical conductivity and MDA content.

### 3.2. Genetic Analysis of Salt Tolerance in SR86

To understand the genetic basis of salt tolerance in SR86, we hybridized SR86 with Nip to obtain an F_2_ population. The seedlings of parental genotypes (SR86 and Nip) and their F_2_ progeny were transplanted into 150 mM NaCl saline-alkaline soil. After 20 days, all Nip plants died, whereas the SR86 plants showed normal growth (Figure 3a). The F_2_ plants showed greater salt tolerance than Nip plants under salt stress (150 mM NaCl) conditions (Figure 3b,d,f). After 10 and 50 days of salt treatment, the F_2_ plants showed partial successive withering. Finally, 30 F_2_ plants reached maturity, as indicated by heading (Figure 3 and Table 1); these 30 plants were considered as the salt-tolerant population.

SR86 was also hybridized with 9311 to obtain an F_2_ population. The seedlings of parents (SR86 and 9311) and F_2_ offspring were transplanted into 150 mM NaCl saline-alkaline soil. All 9311 plants died (Figure 3a). Each F_2_ plant was divided into two seedlings, one of which was transplanted into soil containing no salt, while the other was transplanted into soil containing 150 mM NaCl. After 10 and 50 days of salt treatment, the F_2_ plants showed partial successive withering. Finally, 230 F_2_ plants survived in saline (150 mM NaCl) soil (Figure 3 and Table 1). These results suggest that the salt tolerance of SR86 is a dominant trait.

### 3.3. Salt Tolerance Related QTLs in SR86

To identify salt tolerance related genes in SR86, we selected F_2_ individuals with extreme phenotypes (salt-sensitive and tolerant) from each population (SR86 × Nip and SR86 × 9311) to prepare mixed DNA pools (salt-intolerant and salt-tolerant) for BSA. Four mixed DNA pools were constructed for the SR86 × Nip cross (R01, SR86; R02, Nip; R03, salt-tolerant samples, R04, salt-sensitive samples) and subjected to DNA sequencing. A total of 61.57 Gb clean data were obtained (Q20 ≥ 97.38%, Q30 ≥ 92.98%, GC content = 42.27–42.5%) in all four mixed pools (Appendix A). The sequencing depth of R01 and R02 pools was 29× and 30×, respectively, while that of the R03 and R04 pools was 41× and 43×, respectively (Appendix A). Similarly, four mixed DNA pools were constructed for the SR86 × 9311 cross (SR86; 9311; G1, salt-tolerant samples; G2, salt intolerant samples) and used for DNA sequencing. A total of 41.02 Gb clean data were obtained (Q20 ≥ 96.09%, Q30 ≥ 90.14, GC content = 43.29–43.88%) in all four mixed pools (Appendix A). The sequencing depth of the two parental pools, SR86 and 9311, was 20.53× and 14.79×, respectively, while that of the two offspring pools, G1 and G2, was 26.16× and 28.98×, respectively (Appendix A).

The assay sequences were aligned to the Nip reference genome and we identified approximately 2.14, 0.014, 2.09, and 2.13 million SNPs and more than 45,200, 800, 44,200, and 43,000 Indels, respectively, in four pools (R01–R04) (Appendix A). The △SNP-index between R03 and R04 was calculated and visualized (Figure 4 and Appendix A). Intervals exceeding the threshold of 0.99 were found on chromosome 4 (Figure 4 and Table 1). These areas were considered candidate areas, which corresponded to approximately 5.24 Mb in size and contained 835 genes (with 557 nonsynonymous and 195 frameshift mutations). The results showed that the gene of SR86 related to salt tolerance might be located in a region of 5.24 Mb of chromosome 4. Functional annotation using GO, KEGG, COG, NR, and Swiss-Prot databases revealed that some of the genes in the candidate region encoded salt stress responsive proteins (Figure 5 and Table 1); these genes were investigated further.

When the SR86, 9311, G1, and G2 pooled samples were aligned to the reference genome of 9311, we identified approximately 1.18 million SNPs and more than 32,200 Indels. In order to directly show the distribution of the SNP-index on the chromosome in the offspring, a mapping analysis was carried out. The average value of the SNP-index in each window was calculated to reflect the SNP-index distribution of the offspring and then △ (SNP-index) was calculated (Figure 6). Finally, a 95% confidence level was used as the threshold for screening. The Indel-index was analyzed using the same method as that described for SNP-index analysis. To directly reflect the distribution of the All-index on chromosomes after combining the SNP-index and Indel-index of the offspring, the distribution of the All-index on chromosomes was mapped. The region close to the threshold (the blue line) was selected as a candidate region. Finally, six candidate genes were identified in these regions (Table 2).

### 3.4. Validation of Candidate Gene Expression under Salt Stress by RT-qPCR

To verify whether the candidate genes responded to salt stress, the expression levels of these genes were investigated in SR86, Nip, and 9311 plants by RT-qPCR. The results showed that the expression of *LOC_Os04g03320.1*, *LOC_Os04g03360.1*, and *LOC_Os04g08350.1* was higher in the SR86 plants than in Nip plants and the expression of *LOC_Os04g03320.1* in SR86 increased by 1000-fold within 24 h (Figure 7a,b). This suggests that *LOC_Os04g03320.1* plays a crucial role in the salt tolerance of SR86. Similarly, *BGIOSGA005945*, *BGIOSGA024918*, *BGIOSGA019540*, *BGIOSGA032677*, and *BGIOSGA038294* were expressed to higher levels in SR86 plants than in 9311 plants (Figure 7c,d). However, the expression of *BGIOSGA019540* in SR86 was significantly higher than that in 9311 after 4 h of salt stress treatment (Figure 7c,d), suggesting that *BGIOSGA019540* is critical for the salt tolerance of SR86.

## 4. Discussion

The high salt concentration in soil affects seed germination, seedling growth, and crop yield and poses a serious threat to agricultural production. Therefore, improving the salt tolerance of agriculturally important crops such as rice is of great significance. Rice seedlings are extremely sensitive to salt stress at the three-leaf stage and exhibit a reduced survival rate. Researchers have conducted considerable research on the physiological and molecular mechanisms of salt tolerance at this stage. Under salt stress, lipid peroxidation occurs in plant cells, resulting in the production of reactive oxygen species (ROS), including H_2_O_2_, and a significant increase in the accumulation of MDA [28,29,30]. Additionally, salt stress damages membrane lipids via oxidation, thus increasing the relative conductivity of the plant cells [31]. The accumulation of salt in plant leaves reduces chlorophyll content and decreases photosynthesis [32,33,34]. In previous studies on rice, an increase in the content of organic molecules (such as trehalose, proline, and betaine) improved the salt tolerance of plants [35,36] and an increase in the activity of antioxidant enzymes resulted in the removal of the accumulated products of oxidative damage [37,38]. In the current study, SR86 showed better stability of the cell membrane structure and chlorophyll content, greater accumulation of osmotic substances such as proline, and higher antioxidant enzyme activity under salt stress conditions compared with Nip and 9311. Thus, these physiological parameters may underlie the mechanism of salt tolerance in SR86.

SR86 is a semiwild rice germplasm with strong salt tolerance [20]. Studying the genetic mechanism of salt tolerance and exploring the salt tolerance related genes in SR86 are of great practical significance. However, the analysis of salt tolerance related QTLs in SR86 remains limited. In this study, the phenotypic analysis of SR86 revealed salt-tolerant characteristics, such as normal seedling growth, in saline soil (150 mM NaCl). Additionally, SR86 was hybridized with the *japonica* rice Nip and the *indica* rice 9311, and the salt tolerance of the resulting F_2_ progenies was analyzed to understand the mechanism of salt tolerance in SR86 at the genetic level. Traditional QTL mapping and map-based cloning are limited by several factors and are highly time-consuming. He et al. isolated a major salt tolerance gene over a period of 8 years [39,40]. Transcriptome sequencing led to the identification of a great number of DEGs. With the advent of second-generation sequencing and new sequencing technologies such as BSA-seq, salt-tolerant genes can be isolated from F_2_ populations of salt-tolerant mutants and wild-type hybrids within two years [41]. In recent years, researchers have been able to quickly and accurately map salt tolerance related QTLs and genes by combining QTL-seq with various technologies. Lei et al. identified a candidate gene encoding a C_2_H_2_ zinc finger protein using QTL-seq and RNA-seq [16]. In this study, we used QTL-seq and BSA to quickly locate salt tolerance QTLs on chromosome 4 in the SR86 × Nip F_2_ population. The expression of salt tolerance related candidate genes in SR86 was significantly higher than that in Nip. Among these candidate genes, the expression levels of *LOC_Os04g03360.1* and *LOC_Os04g03320.1* were 100- and 1000-fold higher in SR86 than in Nip. Additionally, six salt tolerance related candidate genes were rapidly located in the F_2_ hybrid population derived from the cross between SR86 (salt-tolerant variety) and 9311 (salt-sensitive variety). Except for *BGIOSGA006170*, all candidate genes showed a significantly higher expression in SR86 than in 9311 plants. The expression level of *BGIOSGA019540*, which encodes a C_2_H_2_ zinc finger protein, was more than 60-fold higher in the SR86 plants than in 9311 plants. Thus, *BGIOSGA019540* may be a key gene for the salt tolerance of SR86 rice and deserves further attention and research.

In previous studies on rice plants at various developmental stages, multiple QTLs related to the salt tolerance were identified on different chromosomes, including chromosome 7 [42], chromosome 5 [43], chromosome 1 [44], and chromosome 6 [45]. Most of the studies on the mapping and cloning of salt tolerance related genes were conducted on seedlings, while research on the other developmental stages is limited. Because the level of salt tolerance in rice varies with the developmental stage, there may be differences without an obvious correlation and genetic basis. Therefore, the breeding of salt-tolerant rice varieties requires QTL analysis or gene mapping in plants at more than one developmental stage, especially the seedling stage and reproductive growth stage, simultaneously. In the current study, 11 salt tolerance related candidate genes were located on chromosomes 2, 4, 5, 7, and 10 using F_2_ plants with extreme phenotypes at the seedling and reproductive stages. The role of these candidate genes in the salt tolerance of SR86 needs to be investigated further.

## 5. Conclusions

In this study, we analyzed the salt tolerance phenotype and genetic effects of SR86. A total of 11 salt tolerance related candidate genes were identified in SR86 × Nip and SR86 × 9311 F_2_ populations at the seedling and reproductive stages using BSA and QTL mapping. RT-qPCR analysis revealed *LOC_Os04g03320.1* and *BGIOSGA019540* as candidate genes controlling salt tolerance in SR86. The QTL identified by the method used in this study could be effectively used in future salt tolerance breeding practices, which has important theoretical significance and application value for revealing the physiological and molecular mechanisms of salt tolerance, making genetic improvements, and breeding salt-tolerant cultivars in rice.

## Figures and Tables

**Figure 1 genes-14-00458-f001:**
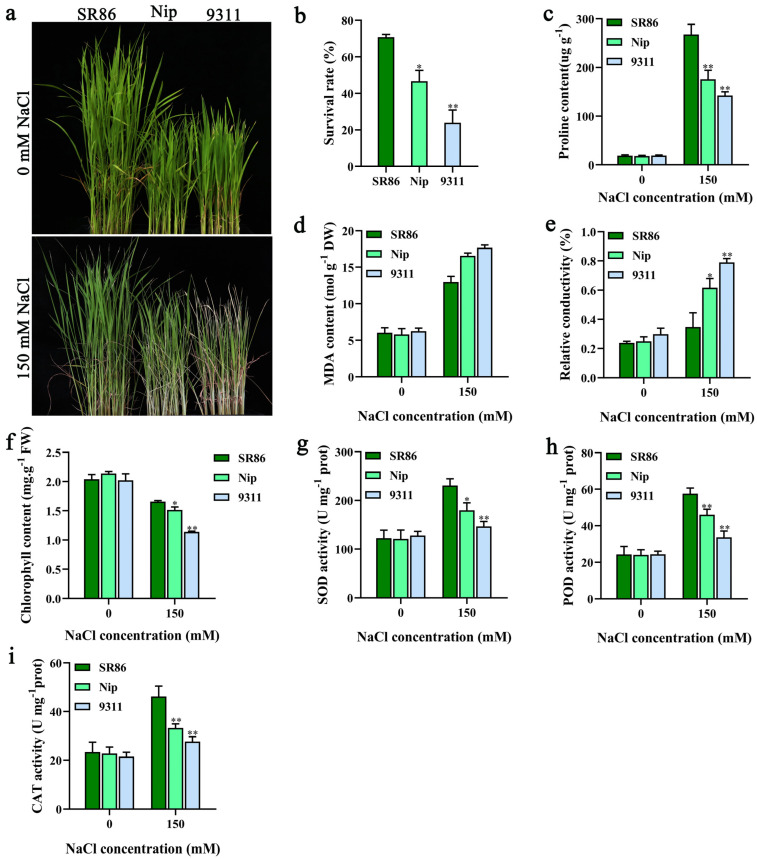
The phenotypic comparison of SR86, Nip, and 9311 plants under salt stress conditions. (**a**−**i**) Phenotype (**a**), survival rate (**b**), proline content (**c**), MDA content (**d**), relative electrical conductivity (**e**), chlorophyll content (**f**), SOD activity (**g**), POD activity (**h**), and CAT activity (**i**) of SR86, Nip, and 9311 plants after treatment with 150 mM NaCl solution for 5 d. The data represent the mean ± SD of three independent replicates. Asterisks indicate statistically significant differences (* *p* < 0.05, ** *p* < 0.01).

**Figure 2 genes-14-00458-f002:**
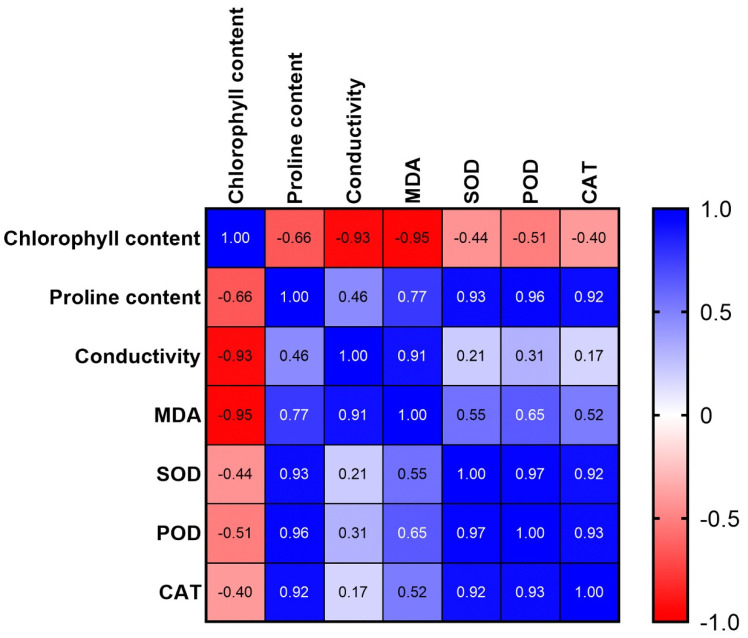
The correlation matrix for related physiological and biochemical indexes.

**Figure 3 genes-14-00458-f003:**
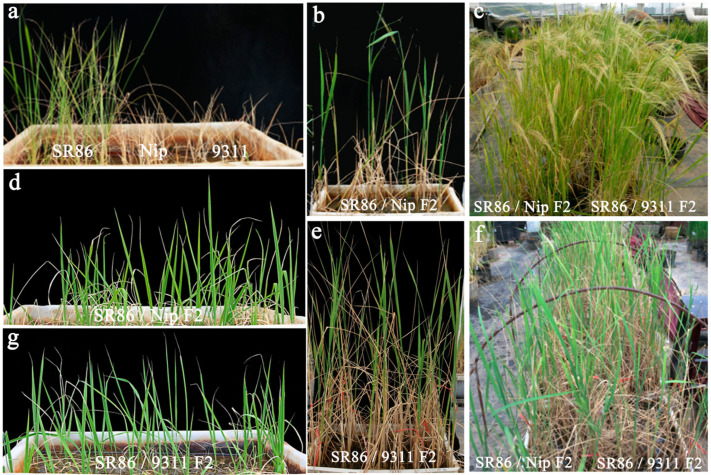
The phenotypic and genetic analyses of SR86 under salt stress. (**a**) Phenotype of SR86, Nip, and 9311 plants exposed to 150 mM NaCl. (**b**) Phenotype of SR86 × Nip F_2_ plants grown in the presence of 150 mM NaCl during the reproductive growth period. (**c**) Phenotype of SR86 × Nip F_2_ plants and SR86 × 9311 F_2_ plants under normal conditions. (**d**) Phenotype of SR86 × Nip F_2_ plants at seedling stage in the presence of 150 mM NaCl. (**e**) Phenotype of SR86 × 9311 F_2_ plants grown in the presence of 150 mM NaCl during the reproductive growth period. (**f**) Phenotype of SR86 × Nip F_2_ plants and SR86 × 9311 F_2_ plants under 150 mM NaCl conditions. (**g**) Phenotype of SR86 × 9311 F_2_ plants at seedling stage in the presence of 150 mM NaCl.

**Figure 4 genes-14-00458-f004:**
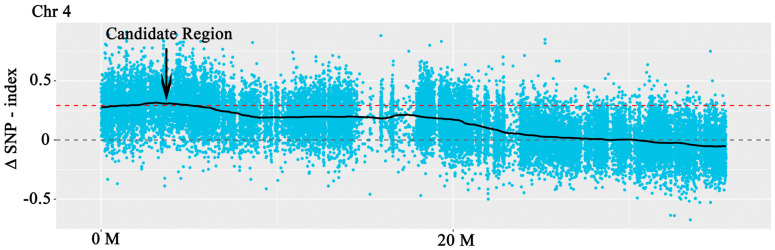
The results of the BSA. Each point in the figure represents the calculated SNP−index or ΔSNP−index value. The black line represents the SNP−index or ΔSNP−index value after fitting and the red dashed line indicates the 99th percentile threshold.

**Figure 5 genes-14-00458-f005:**
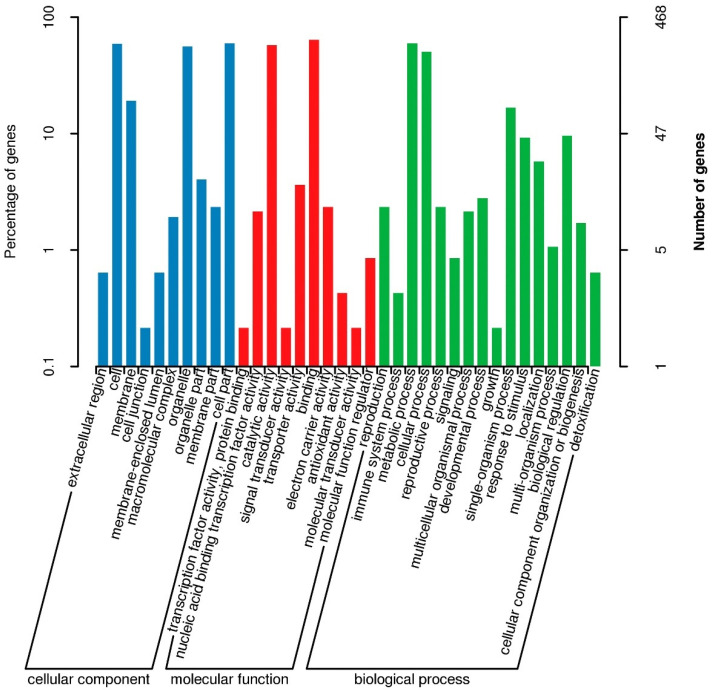
The Gene Ontology (GO) enrichment analysis of candidate regions.

**Figure 6 genes-14-00458-f006:**
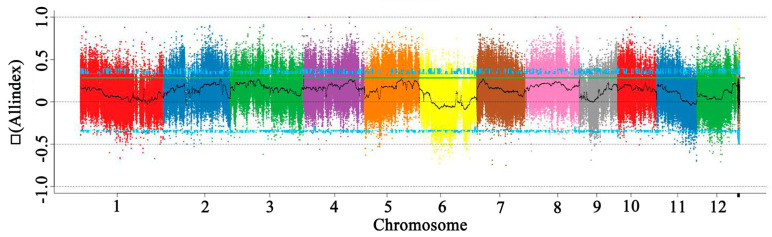
The distribution of Δ(All−index) on the chromosomes of two F_2_ populations. The x−axis indicates chromosome length (Mb) and the y-axis indicates Δ(All−index).

**Figure 7 genes-14-00458-f007:**
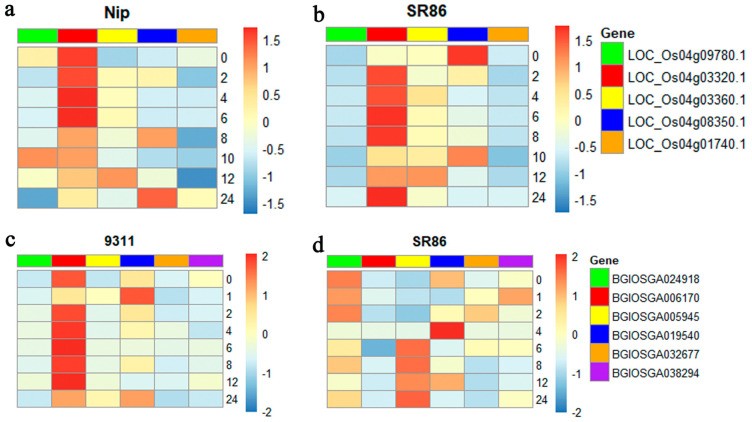
The expression validation of candidate genes in SR86, Nip, and 9311 plants under normal (0 mM NaCl) and salt stress (150 mM NaCl) conditions by RT−qPCR. (**a**) Expression of candidate genes in Nip. (**b**) Expression of candidate genes in SR86. (**c**) Expression of candidate genes in 9311. (**d**) Expression of candidate genes in SR86. Plants were treated with 150 mM NaCl for 24 h. The abscissa indicates the salt concentration and the ordinate shows the relative gene expression level, which was calculated using the 2^−ΔΔCT^ method. Data represent the mean ± SD of three biological replicates.

**Table 1 genes-14-00458-t001:** The mapping and screening of salt tolerance candidate genes.

**Statistics of the Number** **of F_2_ Plants That Survived** **and Reached the Heading Stage After Treatment** **with 150 mM NaCl** **for 10 or 50 Days in Soil**	**F2 Progeny**	**SR86 × Nip**		**SR86 × 9311**	
	NaCl Concentration(mM)	0	150	0	150
	Total No. of Plants	600	600	616	616
	Total No. of Surviving Plants (After 10 d)	600	395	616	591
	Total No. of Surviving Plants (After 50 d)	600	80	616	452
	No. of Plants Showing Heading	600	30	616	230
**Results of** **the Associated Region** **Identified by BSA**	**Chromosome ID**	**Start (bp) ^1^**	**End (bp) ^2^**	**Size (Mb) ^3^**	**Gene Number ^4^**
	Chr4	420,000	5,660,000	5.24	835
	Total	-	-	-	835
**Candidate Gene Information** **in SR86 and Nip plants**	**Locus Name**	**Gene Annotation**			
	LOC_Os04g03320.1	Jacalin-like lectin domain			
	LOC_Os04g08350.4	Pyridoxal-phosphate dependent enzyme			
	LOC_Os04g08350.3	Pyridoxal-phosphate dependent enzyme			
	LOC_Os04g08350.2	Pyridoxal-phosphate dependent enzyme			
	LOC_Os04g08350.1	Pyridoxal-phosphate dependent enzyme			
	LOC_Os04g01740.1	Hsp90 protein			
	LOC_Os04g09780.1	Salt stress response/antifungal			
	LOC_Os04g03360.1	Jacalin-like lectin domain			

^1^ Start: the start position of the associated region. ^2^ End: the end position of the associated region. ^3^ Size: the size of the associated region (Mb). ^4^ Gene number: the number of genes in the associated region.

**Table 2 genes-14-00458-t002:** Candidate Gene Information in SR86 and 9311 plants.

Transcript ID	Mutation Type	Chromosome	Position (bp)	Reference Nucleotide	Alternative Nucleotide
BGIOSGA024918	Nonsynonymous	7	1,043,237	T	A
BGIOSGA024918	Nonsynonymous	7	1,043,246	C	G
BGIOSGA006170	Upstream	2	24,218,884	G	C
BGIOSGA006170	Upstream	2	24,218,891	G	A
BGIOSGA005945	Upstream	2	27,701,048	T	C
BGIOSGA005945	Upstream	2	27,701,050	C	G
BGIOSGA019540	Upstream	5	12,888,795	G	A
BGIOSGA032677	Upstream	10	6,649,316	A	C
BGIOSGA038294	Upstream	CH398734.1	2840	G	A

## Data Availability

All data generated during this study are included in this published article and its Appendix A.

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
