# Peer review of "Identification of Salt Tolerance Related Candidate Genes in ‘Sea Rice 86’ at the Seedling and Reproductive Stages Using QTL-Seq and BSA-Seq"

_genes, 2023, doi:10.3390/genes14020458_

Round 1

Reviewer 1 Report

Dear authors,

I carefully checked your manuscript and found that your paper topic and discussion are suitable for publication in Genes. However, there are some points that should be addressed before publishing this paper.

Please check the following “General” and “Specific” comments to improve the content of your paper before publishing this research.

1- Please provide DOI identifiers for all references cited in this manuscript. Please also make sure that problematic papers or retracted manuscripts have not been cited in your reference list.

2- In the “Funding section” (Line 400): Please make sure the grant numbers for all acknowledged projects were mentioned in this section.

3- The introduction is too long. Please remove all redundant discussions found in the introduction and try to summarize your introduction into three paragraphs: The first paragraph should explain the general facts about the discussed topic, the second one should focus on the core finding of your study and the last one should discuss the major goals of this study. Some lines discussed in the introduction section are general facts about rice and salt stress and are widely discussed in the literature. Please only highlight the top critical points in this section of the paper.

4- Please design a flowchart for your M&M section. This helps academic readers to check all procedures quickly. Please also make sure all applied protocols in this study were cited appropriately.

5- Tables 1-3 (Lines 230 to 237): Please combine these three tables into an integrated table to reduce the number of tables annexed to the main text of the paper. You can design a multi-sectional table for this part of the table.

6- Please supplement raw data for figures 3-5.

7- Please mention which genomic sequencing platform was used in this study. As I checked the paper content, the authors have not mentioned where they sequenced their genomic pools. Please also supplement the FASTQC plots of your sequenced genomes. Additionally, the depth of sequencing is low. Please clarify why these low depths of sequencing were obtained by the sequencing company.

8- Please also mention which assembly, annotation, and mapping tools were used to assemblage and annotate sequenced data. Though the main discussion of this paper focused on data retrieved from the sequencing section, however, enough information on sequencing-related methodology and relevant protocols were not mentioned in the M&M section. Please improve your methodology and add the missing parts to the methods.

9- Which tools were used to conduct gene-ontology analysis? It was unclear how the authors conducted this part of the paper. Please clarify the GO method in the M&M section and try to mention all tools and scripts you used for this purpose. If you use online tools or R packages to generate this plot and previous plots, you should cite the source codes or web servers that you used for this purpose.

10- Please supplement the melt curves and gel electrophoresis of conducted gene expression assays.

11- Please highlight the identified genes based on the indel and SNP index on the rice genome using QTL Cartographer software.

12- Why did the authors only consider 0 and 150mM salt concentrations to conduct gene expression under salt stress? A range of salt concentrations + control groups should be applied to evaluate the expression of target genes.

Reviewer 2 Report

Please provide details for BSA analysis including posterior analysis for DNA pools related to Novogene technology. If there is Linux and or R based protocols are present, please write them all in text.

Correlation matrix and/or PCA analysisshould be carried out for different numeric data such as MDA content, SOD activity, proline content, survival rate etc… Whether there is possitive and/or negative correlation between these different experimental procedures. Graphpad Prism could be used fort his purpose. Also, from LOC_Os04g03320.1 to LOC_Os04g03360.1, each gene could be added to these correlation matrix analysis

PLEASE provide details for qRT-PCR assays. Cycling conditions, mixtures etc..

Please present Fig.6 and Fig.7 as HeatMaps. Graphpad 9.0 or http://www.heatmapper.ca/ could be used for this purpose.

Please provide mean/max/min Cp values for target genes used in qRT-PCR. Moreover, E values and melting scores should be written in text.

Please provide references with the publication year of 2021 and 2022 throughout the text.

Round 2

Reviewer 1 Report

Dear authors,

I have no further comments on this paper. The respected authors carefully addressed my comments and the current draft is suitable for publication. My final decision on this manuscript is "Acceptance" in the present form. 

Regards.